# Single ion qubit with estimated coherence time exceeding one hour

Pengfei Wang [1✉], Chun-Yang Luan [1], Mu Qiao[1], Mark Um[1], Junhua Zhang[1,2], Ye Wang [1,3], Xiao Yuan [4,5], Mile Gu[6,7,8], Jingning Zhang [9] & Kihwan Kim [1✉]

Realizing a long coherence time quantum memory is a major challenge of current quantum technology. Until now, the longest coherence-time of a single qubit was reported as 660 s in a single $^{171}Yb^+$ ion-qubit through the technical developments of sympathetic cooling and dynamical decoupling pulses, which addressed heating-induced detection inefficiency and magnetic field fluctuations. However, it was not clear what prohibited further enhancement. Here, we identify and suppress the limiting factors, which are the remaining magnetic-field fluctuations, frequency instability and leakage of the microwave reference-oscillator. Then, we observe the coherence time of around 5500 s for the $^{171}Yb^+$ ion-qubit, which is the time constant of the exponential decay fit from the measurements up to 960 s. We also systematically study the decoherence process of the quantum memory by using quantum process tomography and analyze the results by applying recently developed resource theories of quantum memory and coherence. Our experimental demonstration will accelerate practical applications of quantum memories for various quantum information processing, especially in the noisy-intermediate-scale quantum regime.

[1] Center for Quantum Information, Institute for Interdisciplinary Information Sciences, Tsinghua University, 100084 Beijing, China. [2] Shenzhen Institute for Quantum Science and Engineering, and Department of Physics, Southern University of Science and Technology, 518055 Shenzhen, P. R. China. [3] Department of Electrical and Computer Engineering, Duke University, Durham, NC 27708, USA. [4] Stanford Institute for Theoretical Physics, Stanford University, Stanford, CA 94305, USA. [5] Center on Frontiers of Computing Studies, Department of Computer Science, Peking University, Beijing 100871, China. [6] Centre for Quantum Technologies, National University of Singapore, Singapore 117543, Singapore. [7] School of Mathematical and Physical Sciences, Nanyang Technological University, Singapore 637371, Singapore. [8] Complexity Institute, Nanyang Technological University, Singapore 637335, Singapore. [9] Beijing Academy of Quantum Information Sciences, 100193 Beijing, China. ✉email: wpf16@mails.tsinghua.edu.cn; kimkihwan@mail.tsinghua.edu.cn

Quantum coherence is a vital component for scalable quantum computation[1–3], quantum metrology[4,5], and quantum communication[6–10]. In practice, decoherence, loss of coherence in the computational basis, in the quantum system comes from the coupling with the surrounding environment and fluctuations of control parameters in quantum operations, which can lead to the infidelity of quantum-information processing, the low sensitivity of quantum sensors, and the inefficiency of quantum repeater based protocols in quantum communication networks. Limited coherence time may also undermine quantum-information applications such as quantum money[11,12]. It is thus of practical importance to have a stable quantum memory with a long-coherence time.

Numerous experimental attempts have been made to enhance the coherence time of quantum memory in a variety of quantum systems. With ensembles of trapped ions and nuclear spins in a solid, coherence time of 10 min[13,14], and 40 min at room temperature[15,16] and a few hours at 4 K[17] have been reported, respectively. For a single qubit quantum memory, which is the essential building-block for quantum computers[18,19] and quantum repeaters[20,21], records of coherence time have been reported to the time scale of a minute in trapped ion qubit[22–25]. For the coherence time of a minute, the limitation mainly came from the qubit-detection inefficiency[25–27] due to the motional heating of qubit-ions without Doppler laser-cooling. The problem was addressed by sympathetic cooling by other species of ion, which allowed further improvements of coherence time to over 10 min with the support of dynamical decoupling[28]. While the fundamental limit is far beyond 10 min; however, it remains a major technological challenge to further enhance the quality of a trapped-ion quantum memory.

Here we address this challenge by improving the coherence time of a $^{171}Yb^+$ ion-qubit memory from 10 min to over one hour. This is achieved by identifying and suppressing the three dominant error sources: magnetic-field fluctuation, the phase noise of the local oscillator, and microwave leakage for qubit operation. Furthermore, with the capability of full control on a single qubit, we systematically study the decoherence process of the quantum memory by quantum process tomography. Typically, the decoherence process has been characterized by the coherence time $T_2$ at which the Ramsey contrast, corresponding to the size of the off-diagonal entry in the qubit density-matrix, decays to $1/e$[13–17,28]. We experimentally study the decoherence dynamics by relevant quantum channels of depolarization and dephasing, which allows us to use recently developed coherence quantifiers[29–31]. We also use our data to study recently developed resource theories of quantum memory and coherence, such as the robustness of quantum memory (RQM) that quantifies how well a memory preserves quantum information[32] and relative entropy of coherence (REC) that quantifies how much coherence is maintained in the state.

## Results

**Two species of atomic ions.** In our experiment, we load one $^{171}Yb^+$ ion and one $^{138}Ba^+$ion in a four-rod Paul trap as shown in Fig. 1a. Two hyperfine levels of the $^{171}Yb^+$ ion in the $S_{1/2}$ manifold are used to encode the qubit with $\{|0\rangle \equiv |F = 0, m_F = 0\rangle, |1\rangle \equiv |F = 1, m_F = 0\rangle\}$ and a frequency difference of $12642812118 + 310.8B^2$ Hz, where $B$ is the magnetic field in Gauss. As a sympathetic cooling ion, $^{138}Ba^+$ is used since it has a similar atomic mass with $^{171}Yb^+$, which can be used for efficient cooling. We apply Doppler-cooling laser beams on the $^{138}Ba^+$ ion all the time, which provides continuous cooling for the whole system. In this way, we can measure the final state of the $^{171}Yb$

$^+$qubit by standard fluorescence detection technique without losing any detection fidelity[25–27].

**Suppression of ambient magnetic field.** We suppress the ambient noise of the magnetic field by installing a magnetic-field shielding with a permanent magnet[33]. We enclose our main vacuum chamber that contains the Paul trap with a two-layer of μ-metal shielding as shown in Fig. 1a. By using a fluxgate meter, we observe more than 40 dB attenuation at 50 Hz inside the shielding, which is the main frequency of noise in the lab due to the AC power-line. To generate stable magnetic field of 5.8 G, we replace coils with a $Sm_2Co_{17}$ permanent magnet, which has a temperature dependence of −0.03%/K[33]. The magnetic field strength can be adjusted by changing the position of the magnet from the location of ions. After these modifications, we observe the coherence time of the field-sensitive Zeeman qubit is increased to more than 30 ms. We study the noise spectrum by dynamical decoupling sequences[34,35] and observe that noise of 50 Hz and 150 Hz are below 16 μG and 32 μG, respectively.

**Improvements of microwave frequency stability.** We perform coherent manipulation of the qubit by applying a resonant microwave. Qubit coherence is typically measured by the contrast of Ramsey fringe, which requires control and interrogation of the system by a local oscillator that can bring in phase noise[36,37]. In our case, this part of the noise is determined by the microwave signal generator and its reference. For microwave signal, phase noise in the low-frequency regime is mainly determined by those of the reference signal. We use a crystal oscillator as the reference, which has an order-of-magnitude smaller Allan variance at 1 s observation time than our previous Rb clock oscillator[28].

**Suppression of microwave power-leakage.** We also find that leakage of the microwave can introduce relaxation of the qubit memory. We include a microwave switch after amplifier as shown in Fig. 1b, which reduces the leakage by over 70 dB. In total, we suppress the microwave output by 164 dB after turning off all the switches. With π pulse duration of 175 μs, the effect of leakage is negligible for 0.4 s pulse interval time, which would be further suppressed by dynamical decoupling pulses. At the same time, we also use AOMs, EOPP and a mechanical shutter to suppress the leakage of $^{171}Yb^+$ ion resonant laser beams as same as Ref. [28].

**Dynamical decoupling pulse sequence.** We measure the coherence time of the $^{171}Yb^+$ ion qubit by observing the dependence of Ramsey contrasts on the storage time. The experimental sequence is shown in Fig. 2. As discussed above, cooling laser beams for $^{138}Ba^+$are applied during the whole sequence. We initialize the state of the $^{171}Yb^+$ ion qubit to $|0\rangle$ by the standard optical pumping technique, apply the π/2-Ramsey pulses, and detect the probability in $|1\rangle$ state by the standard state-dependent fluorescence method. In the Ramsey measurement, we observe the coherence time of 1.6 s (see "Methods" for the details). We note that we have a detection efficiency of 98.6%, which is corrected by the calibrated error magnitude with the uncorrelated error assumption as shown in Ref. [38].

To enhance coherence time, we first apply a spin-echo pulse that uses a single π pulse to compensate low-frequency noise. We observe the coherence time is improved to 11.1 s with the single spin-echo pulse (see "Methods" for the details). We then apply the dynamical decoupling scheme[16,17,28,34,35,39,40], which contains multiple of spin-echo pulses. Performance of dynamic

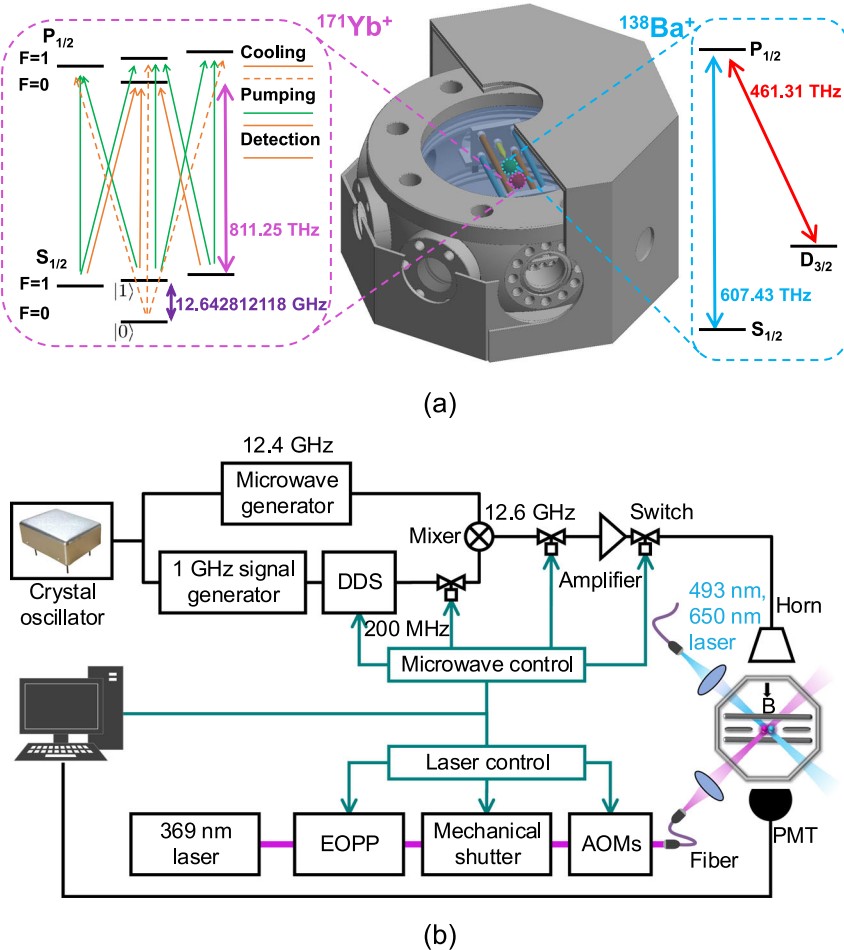

(a)

(b)

**Fig. 1 Experimental setup. a** Energy levels of $^{171}$Yb$^+$ and $^{138}$Ba$^+$ ion and cutaway view of the $\mu$-metal shielding enclosing octagon chamber. The shielding has ten holes, where two holes for connection of the vacuum pump and helical resonator and the other eight holes with diameter from 20 to 40 mm for the access of laser beams, microwave, and imaging system. **b** The schematic diagram for the control of microwave and laser beams. We use a crystal oscillator (SIMAKE SMK3627OCHFM OCXO) to reference the microwave generator and Direct Digital Synthesize (DDS) through a 1 GHz signal generator. The microwave of 12.6 GHz is generated by mixing 200 MHz signal from DDS and 12.4 GHz from the microwave generator, which is amplified and applied to ions through a horn. All three microwave switches are used to reduce microwave leakage. For 369 nm laser beams, we use acousto-optic modulators (AOMs) to generate basic operating lasers. We use Electro-Optic pulse picker (EOPP), mechanical shutter and single-mode fiber to reduce laser leakage. The magnetic-field direction is in the radial direction. We detect the qubit state with a photomultiplier tube (PMT).

decoupling pulses is described by the filter function $\widetilde{y}(\omega, T) = \frac{1}{\omega} \sum_{j=0}^{T/\tau} (-1)^j \left( e^{i\omega t_j} - e^{i\omega t_{j+1}} \right)$, with $t_0 = 0$, $t_{(T/\tau)+1} = T$, $t_j = (j - 0.5)\tau$ when $1 \leq j \leq T/\tau$, and $\tau$ is the interval of pulses. Then Ramsey fringe contrast[34] is $W(T) = e^{-\frac{2}{\pi} \int_o^\infty S(\omega) |\widetilde{y}(\omega, T)|^2 d\omega}$ with $S(\omega)$ being the noise spectrum density. In our experiment, we use KDD$_{xy}$(Knill dynamical decoupling)[17,28,40] pulses, where all the pulses are equally spaced and have periodic phases as shown in Fig. 2. The filter function of the KDD$_{xy}$ pulses has a peak at the frequency of $\omega = \frac{\pi}{\tau}$. Most of the noise is suppressed except the part with frequencies around the peak, which is instead amplified. When the total time $T$ is fixed, the position of the peak is determined by the pulse interval, which can be optimized depending on the noise spectrum. After comparing different parameters, we choose 0.4 s as the pulse interval, which leads to the peak of the filter function at $2\pi \times 1.25$ Hz.

**Resulting coherence time**. With different initial states, we show the time dependence of the Ramsey contrast up to 960 s in Fig. 3. By assuming exponential decay of the Ramsey contrast,

we find the coherence time of states $|0\rangle$ and $|1\rangle$ to be 16000 ± 3200 s. Other four superposition states ($\phi = 0$, $\frac{\pi}{2}$, $\pi$, and $\frac{3\pi}{2}$ shown in the legends of Fig. 3) have a coherence time of 5500 ± 670 s. Both of the uncertainties are from fitting errors. As shown in the inset of Fig. 3, the coherence time is increased by an order-of-magnitude compared to the previous state-of-the-art result[28].

**Experimental study of decoherence process**. We further analyze the decoherence process by performing quantum process tomography, which completely characterizes unknown dynamics of a quantum system, at different storage time following Refs. [41,42]. The procedure of quantum process tomography is as follows. For a quantum process $\varepsilon$, we consider its process $\chi$ matrix, which is defined by $\varepsilon(\rho) = \sum_{mn} \chi_{mn} \hat{E}_m \rho \hat{E}_n^\dagger$ with $\hat{E}_m \in \{\hat{I}, \hat{X}, \hat{Y}, \hat{Z}\}$[42]. We measure the $\chi$ matrix of our single ion-qubit memory by preparing four different input states $|0\rangle$, $|1\rangle$, $(|0\rangle + |1\rangle)/\sqrt{2}$, $(|0\rangle + i|1\rangle)/\sqrt{2}$, applying the memory, and finally measuring the output states with four measurements $I$, $X$, $Y$ and $Z$. We use the maximum likelihood method to reconstruct the process matrix[41]. We observe

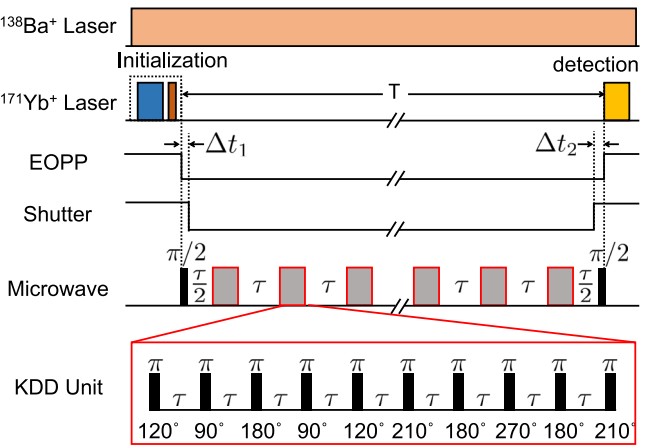

**Fig. 2 Experimental sequence.** Cooling laser beams for the $^{138}$Ba$^+$ ion are applied during the whole sequence. For $^{171}$Yb$^+$ ion, we first initialize the qubit and then start to apply the microwave pulses. All the KDD$_{xy}$(Knill dynamical decoupling) pulses are inserted between two $\pi/2$ pulses of Ramsey sequence. Blue and brown blocks represent Doppler cooling and optical pumping pulses for $^{171}$Yb$^+$ ion. EOPP and shutter are closed after state initialization and opened before state readout, where the time delays between them are shown as $\Delta t \approx 10$ ms, which is mainly caused by the limited speed of the mechanical shutter. Gray blocks represent KDD$_{xy}$units. $T$ is the total measurement time, and $\tau$ is the interval of pulses. Each KDD$_{xy}$ unit has ten $\pi$ pulses, where the first and the second five pulses represent $\sigma_x$- and $\sigma_y$-rotation, respectively. Therefore, the second five pulses have 90° phase shift from the first five. We choose the total number of KDD$_{xy}$units even to make sure all the KDD$_{xy}$ pulses are identity operation in the ideal case. In the end, we use a detection laser pulse to measure the qubit state.

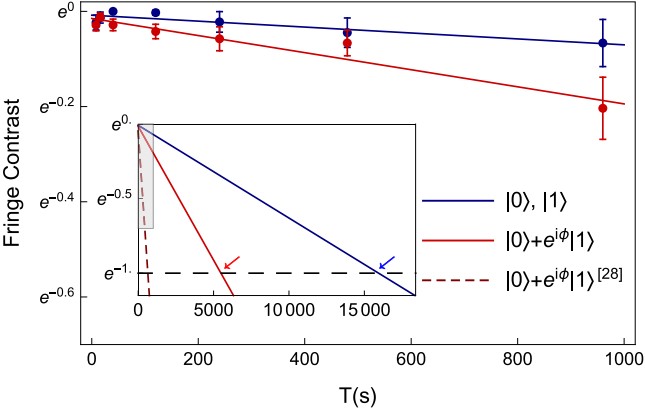

**Fig. 3 Blue points are from the initial states of $|0\rangle$ and $|1\rangle$, and red points are from $|0\rangle + |1\rangle$, $|0\rangle + i|1\rangle$, $|0\rangle - |1\rangle$, and $|0\rangle - i|1\rangle$, where $\phi = 0$, $\frac{\pi}{2}$, $\pi$, and $\frac{3\pi}{2}$, respectively.** Error bars are standard deviations. Each initial state at each data point repeats 30 to 100 times. The solid lines are the fitting results by the exponential decay function. Inset shows extrapolations of fits in a longer time range. The shadow indicates the enlarged area in the figure. The red-dashed line indicates the previous result of superposition states[28]. The black-dashed line indicates the 1/$e$ threshold. The red and blue arrows indicate times when threshold are reached.

the time dependence of the process matrix as shown in Fig. 4a. The ideal quantum memory process is described by $\chi^{id}_{mn} = \delta_{m,1}\delta_{n,1}$. With the experimentally measured process matrix $\chi^{exp}$, we can obtain the process fidelity $F_p = \text{Tr}(\chi^{id}\chi^{exp}) = \chi^{exp}_{11}$. The infidelity mainly comes from the dephasing and depolarization effects. The

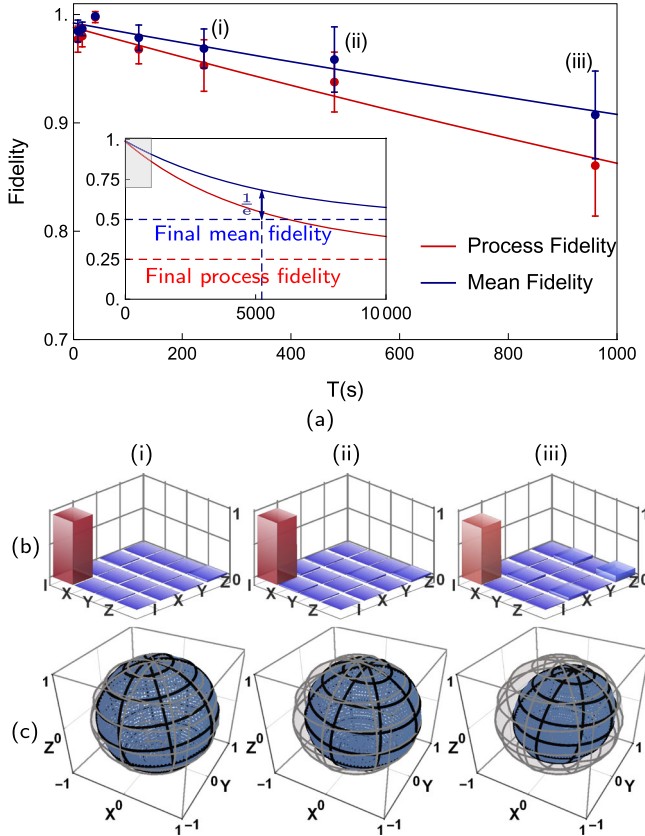

**Fig. 4 Results of quantum process tomography. a** Red and blue points represent process and mean fidelities, respectively. Error bars are standard deviations. The red line is the fitting result of Eq. (1). The blue line is the fitting result of the exponential decay function. Inset shows extrapolations of fits in a longer time range. The shadow indicates the enlarged area in the figure. The red and blue dashed horizontal lines indicate the process fidelity and mean fidelity of the final state, where the system lost all the quantum information. The blue vertical line indicates the time point when mean fidelity decays to 1/$e$ threshold. (**b**), The real part of the process matrix after a storage time of (i): 4 min, (ii): 8 min, and (iii): 16 min. The largest diagonal element of the process matrix is the identity operation part, $\chi^{exp}_{11}$, which is the process fidelity $F_p$. (c), State evolution represented in the Bloch sphere after a storage time of (i): 4 min, (ii): 8 min, and (iii): 16 min. Gray meshed spheres represent the initial pure states, which form the Bloch sphere. And blue spheres represent the output states after corresponding storage time, which are the same as the input state at $T = 0$ but shrink into the Bloch sphere later and transition to a dot in the center for $T \gg \min(T_1, T_2)$. Given the input state, the corresponding output state is calculated by the process matrix $\chi^{exp}$.

process with these two noises can be described by the following matrix as

$$\begin{bmatrix} \frac{1+2e^{-t/T_2}+e^{-t/T_1}}{4} & 0 & 0 & 0 \\ 0 & \frac{1-e^{-t/T_1}}{4} & 0 & 0 \\ 0 & 0 & \frac{1-e^{-t/T_1}}{4} & 0 \\ 0 & 0 & 0 & \frac{1-2e^{-t/T_2}+e^{-t/T_1}}{4} \end{bmatrix}, \quad (1)$$

where $T_1$ and $T_2$ are depolarizing and total dephasing time, respectively[43,44]. The process matrix describes a quantum memory with full coherence at $T = 0$ but which has transitioned to a fully mixed state for $T \gg \min(T_1, T_2)$. By fitting the experimental process tomography results with the above process matrix of Eq. (1), we obtain $T_1 = 12000 \pm 2200$ s and $T_2 = 4200 \pm 580$ s (see

Methods for the details). We also plot the model of Eq. (1) and the experimental data in Fig. 4a.

From experimental quantum process tomography, the performance of the quantum memory on arbitrary quantum states can be accurately estimated, which can be simplified as the mean fidelity, $F_{\mathrm{mean}} = \langle \mathrm{Tr}(\rho \varepsilon(\rho)) \rangle_\rho$, which is the averaged output fidelity with all possible input states $\rho$[45–47]. The mean fidelity is a function of wait time T since the process matrix of quantum memory is different depending on wait time $T$. We use the Monte Carlo method to get the mean fidelity with $10^5$ different input states, generated by uniformly sampled random unitary operations according to the Haar measure[48]. As shown in Fig. 4a, we obtain the coherence time, the time constant of fitted exponential decay function, 5200 ± 500 s for the mean fidelities. We note that within the error bar, this coherence time is consistent with that of a simple estimation of the mean fidelity from the formula of $F_{\mathrm{mean}} = (2F_{\mathrm{p}} + 1)/3$[45], where it provides 5600 ± 650 s.

**Benchmark of quantum memory and quantum coherence.** Recently due to the fundamental importance of quantum coherence, there have been serious developments of rigorous theories of quantum coherence and quantum memory as a physical resource. In our manuscript, we relate our experimental results with up-to-date resource theories of quantum coherence and quantum memory such as REC and RQM, respectively.

The REC is a distance-based coherence quantifier, which is suggested as a gold standard measure[45]. The REC can be interpreted as the minimal amount of noise required for making a quantum state fully decohere[31]. The REC has the same formula with distillable coherence, which has an analogy to the distillable entanglement, a standard widely using entanglement quantifier. The distillable coherence is the optimal number of maximally coherent single-qubit states that can be obtained in a given qubit state through incoherent operations and fulfills all the requirements as a proper coherence quantifier[31]. The formal definition of the REC[30] is written as $C(\rho) = S(\Delta(\rho)) - S(\rho)$, with $\Delta(\rho) = \sum_i \langle i|\rho|i\rangle |i\rangle\langle i|$, $\{|i\rangle\}$ being the computational basis, and $S(\rho) = -\mathrm{Tr}(\rho \log_2 \rho)$ being the Von Neumann entropy.

In our analysis, we use the ratio of the REC between the output state and the input state instead of directly using the REC because each input state has a different value of the REC. Based on the process matrix $\chi^{\mathrm{exp}}$, we numerically calculate the ratio of the REC. We study the time dependence of the mean ratio of the REC $C'_{\mathrm{mean}} = \langle C(\varepsilon(\rho))/C(\rho) \rangle_\rho$, where we average over $10^5$ random input states. Note that we only consider states with REC larger than 0.01. As shown in Fig. 5, the mean REC ratio decays to $1/e$ after 3500 ± 1100 s by the exponential fitting. The relatively short duration and the large fluctuation of the results mainly stem from stringent condition and sensitivity of the REC to small errors in the process matrix.

The RQM, which was introduced by Ref. [32], quantifies how well the memory preserves quantum information that includes coherence. Here the quantum memory, which stores a quantum state for later retrieval, is considered as a channel that maps an input state to an output state. Ideally, it should be an identity channel. The quantifier of RQM is developed based on the approach that considers the quantum memories as a resource and provides a means to benchmark quantum memories. Basically, the higher the RQM is, the more noise the quantum memory can sustain before it is unable to preserve quantum information. In contrast, a classical memory that cannot preserve quantum information is characterized as a measure-and-prepare (MP) memory that destroys the input state by measurement, and stores only the classical measurement result.

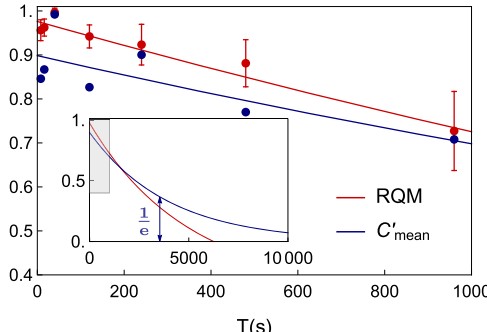

**Fig. 5 Benchmark of quantum memory and coherence.** Red and blue points are data of the robustness of quantum memory (RQM) and the mean ratio of the relative entropy of coherence (REC), respectively. Error bars are standard deviations. The red line is the theoretical result of the RQM calculated from the process matrix of Eq. (1) and the blue line is the exponential fitting result of the mean ratio of the REC. Inset shows extrapolations of fits in a longer time range. The shadow indicates the enlarged area in the figure. The blue vertical line indicates the time point when the mean REC ratio decays to $1/e$.

The RQM is defined as the least portion of the classical memory that needs to be mixed with the quantum memory so that the resultant mixture belongs to MP memory, which is formally written as $R(\mathcal{N}) = \min_{\mathcal{M} \in \mathcal{F}} \left\{ s \geq 0 \,|\, \frac{\mathcal{N} + s\mathcal{M}}{s+1} \in \mathcal{F} \right\}$, where $\mathcal{N}$ is the quantum memory of interest, $\mathcal{M}$ is a classical memory that is in the set of MP memories $\mathcal{F}$, and $s$ is the amount of mixture of the quantum memory $\mathcal{N}$ with the classical memory $\mathcal{M}$. The RQM is the minimum value of $s$ to make the mixed memory in $\mathcal{F}$. We note that the RQMs of all classical memories are zero since the MP memories cannot maintain quantum information. We obtain the RQM from the experimental process matrix. In general, the $R(\mathcal{N})$ can be found by a numerical search of the minimum $s$. Assuming off-diagonal elements in the process matrix are negligible, the RQM can be simplified to $\max\{2F_{\mathrm{p}} - 1, 0\}$ for qubit quantum memories. In our experimental process tomography, no noticeable difference is observed between the numerical search and the simplified formula. As shown in Fig. 5, the RQM of our system lasts 6300 s before it decays to zero by the exponential fitting.

## Discussion

In conclusion, we report a trapped-ion based single qubit quantum memory with over one hour coherence time, an order-of-magnitude enhancement compared to the state-of-the-art record[28]. The quantum memory with the long-coherence time will accelerate the development of scalable quantum computation[3,49,50], long-distance quantum communication[9,51], high-precision quantum metrology[4,5], and quantum money[11,12], in particular, in the near-term noise-intermediate-scale quantum regime where there will be no quantum error correction. Our research can be also extended to realize a general-purpose quantum memory that contains multiple qubits capable of individual storage and retrieval of quantum information at any required time with further enhancement of coherence time and increase of the number of individually controllable qubits.

Further enhancement of the coherence time to day level ($\approx 10^5$ s) may be achievable by improving the stability of the classical oscillator and magnetic-field fluctuation as shown in Fig. 6 (see also "Methods"). To reach the ultimate coherence time limited by the lifetime of the excited hyperfine state that is expected to be thousands of years to our estimation (see Methods), we need to suppress the hopping of ions, decoherence from scatting of $^{138}\mathrm{Ba}^+$ lasers,

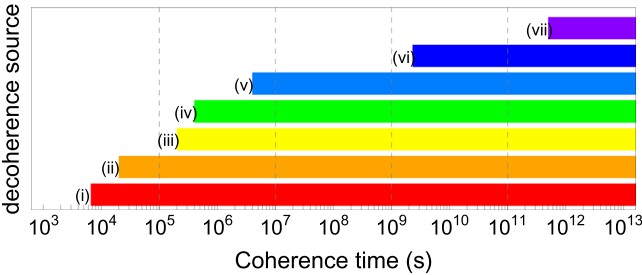

**Fig. 6 Expected limitations of coherence time.** The left boundaries of different color-bars indicate expected limitations caused by corresponding decoherence sources as follows: (i) Phase noise of local oscillator; (ii) Magnetic-field fluctuation; (iii) Ion hopping; (iv) Scatting of $^{138}Ba^+$ lasers; (v) Leakage of the microwave; (vi) Collision of background gas; (vii) Lifetime of hyperfine ground-state. Currently, the coherence time is mainly limited by the phase noise of the local oscillator and the ultimate coherence time limited by the lifetime of the hyperfine state is estimates as around $5 \times 10^{11}$ s.

leakage of the microwave, and collision of the background gas. Microwave leakage can be simply addressed by adding switches. The other sources of decoherence are related to the background gas collisions. The collisions cause hopping of ions, which introduces frequency shift from different magnetic-field strengths between two positions and collision frequency shift due to change of motional distribution and phase of atomic superposition[52]. The background gas collisions can be significantly suppressed by locating the ion trap system in a cryostat environment[53], which naturally suppresses the hopping rates and collision-induced shift. No hopping allows us to shed the cooling laser beams only on the $^{138}Ba^+$ ion, which eliminates the scatting-induced decoherence of the ion qubit (see "Methods").

Our work can be extended to the general purpose of quantum memory, quantum money for example, that requires a large number of qubits by using a long ion-chain in a trap with an individual addressing system. The necessary technical improvement for such quantum memory is to eliminate the hopping problem because hopping ruins the individual tracking of the quantum memory. The hopping problem in the long-linear chain can be also suppressed by a cryostat ion trap as discussed above. We also notice that in the long ion-chain, the micromotion induces inefficiency of state-detection[54]. Individual compensation of the micromotions can be achieved by a sophisticated trap with the capability of local-field control.

## Methods

**Expected limitations of coherence time.** The expected limitations of coherence time caused by different decoherence sources are summarized in Fig. 6. We note that in the analysis, we do not consider the imperfection of the $KDD_{xy}$ pulses because we find the $KDD_{xy}$ sequence is robust against the typical errors as flip-angle error and frequency-offset errors even at the levels of errors in our system[40]. For the flip-angle error of $10^{-2}$ and the frequency-offset error of 100 Hz, around $2 \times 10^{10}$ pulses and $3 \times 10^{10}$ can be applied before the output results decay to $1/e$, respectively, which correspond to $0.8 \times 10^{10}$ s and $1.2 \times 10^{10}$ s, respectively, for our choice of the gap-time, 0.4 s.

(i) Phase noise of local oscillator: the new frequency reference for local oscillator has an order-of-magnitude smaller Allan variance $\sigma(\tau_0)^2$ at $\tau_0 = 1$ s than that of previous one in Ref. [28], which indicates an order-of-magnitude smaller phase-noises spectrum density $S_{LO}(\omega)$, assuming the shape of $S_{LO}(\omega)$ is the same for both references. It is because of the relation between Allan variance $\sigma(\tau_0)^2$ and phase-noise spectrum density $S_{LO}(\omega)$ is $\sigma(\tau_0)^2 = \frac{1}{\pi} \int_0^\infty S_{LO}(\omega) \sin^4(\frac{\tau_0}{2}\omega) d\omega$[55]. With the order-of-magnitude smaller $S_{LO}(\omega)$, the Ramsey fringe contrast[34] $W(T) = e^{-\frac{2}{\pi}\int_o^\infty S_{LO}(\omega)\left|\widetilde{\gamma}(\omega,T)\right|^2 d\omega}$ will also takes an order-of-magnitude longer time to reach $1/e$. Therefore, the current an order of magnitude enhancement of coherence time is mainly limited by the phase noise of local oscillator.

(ii) Magnetic-field fluctuation: magnetic-field noise is suppressed by shielding and permanent magnet. The comparison of magnetic-field fluctuation before and

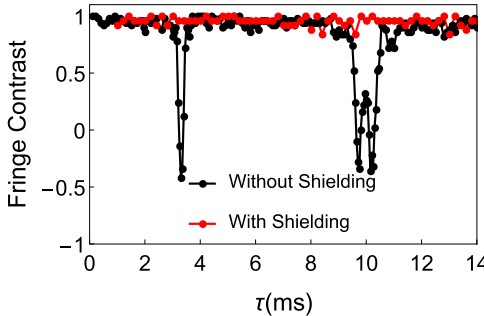

**Fig. 7 Suppression of magnetic-field noise.** To check the noise suppression of magnetic-field shielding, we use 31 CPMG (Carr, Purcell, Meiboom, and Gill) pulses to accumulate the AC magnetic-field noise[28]. The figure shows Ramsey contrast as a function of the inter-DD pulse spacing $\tau$. Black and red points represent data without and with shielding and permanent magnet[28], respectively. Before the improvement of magnetic field stability, there are two dips at $\tau = 3.3$ ms and $\tau = 10$ ms which correspond to 150 Hz and 50 Hz noise, respectably, which disappeared after the improvement. We further increase the CPMG pulses number to 190, and get fringe contrasts of 0.97 and 0.98 at $\tau = 10$ ms and 3.3 ms, respectively. This indicates that the level of the noise at 50 Hz and 150 Hz are below 16 $\mu$G and 32 $\mu$G, respectively.

after the suppression is shown in Fig. 7. The coherence time of the Zeeman state is improved by around 30 times improvement after magnetic-field noise suppression similar to that in Ref. [33]. Therefore, we expect the limitation of the coherence time of the clock-state qubit due to the magnetic-field fluctuation is increased by 30 times, which is around $2 \times 10^4$ s.

(iii) Ion hopping: hopping of the ions between two positions that have the qubit-frequency difference of 0.22 Hz (60 $\mu$G difference) occurs about every 10 min. The estimated infidelity of a superposition state due to the alternating frequency changes from the ion hopping is around $2.7 \times 10^{-3}$ per hopping. Assuming the infidelity increases exponentially with the number of hopping, the limitation of coherence time due to the ion hopping is expected to be around $2 \times 10^5$ s $(= 10$ min $/(2.7 \times 10^{-3}))$. We estimate the infidelity per hopping as follows. Since a small amount of constant frequency shift almost does not introduce infidelity due to the $KDD_{xy}$ sequences, we ignore the no-hopping period. When hopping occurs, the effect of frequency shift cannot be compensated by the dynamical decoupling pulses, which introduces the infidelity of the state. We assume in one $KDD_{xy}$ unit, hopping occurs at most once with a uniform distribution of time, which is reasonable since the duration of one $KDD_{xy}$ unit (4 s) is much shorter than the period of the hopping (10 min). Finally, we average out the infidelities at different occurring time of hopping.

(iv) Scatting of $^{138}Ba^+$ lasers: we estimate the spontaneous emission rate of the $^{171}Yb^+$ ion assuming the cooling laser beams (493 nm and 650 nm) of the $^{138}Ba^+$ ion are entirely applied to the $^{171}Yb^+$ ion. The spontaneous emission rate of the dipole transition of the $^{171}Yb^+$ ion is written as[28,56–58]

$$\Gamma_{spon} = \frac{\gamma g^2}{6}\left(\frac{1}{\Delta_{D1}^2} + \frac{2}{(\Delta_{FS} - \Delta_{D1})^2}\right), \qquad (2)$$

where $\gamma \approx 2\pi \times 20$ MHz is the spontaneous emission rate from the $^2P$ states, $g = \frac{\gamma}{2}\sqrt{I/(2I_{sat})}$, $\Delta_{HF} = 2\pi \times 12.6$ GHz, $\Delta_{FS} = 2\pi \times 100$ THz. For 493 nm laser, power $P = 35$ $\mu$W, beam waist $\omega = 31.4$ $\mu$m, $I_{493} = 21.8I_{sat}$, $\Delta_{D1} = 2\pi \times 203.8$ THz, then we get a scattering rate of $1.09 \times 10^{-6}$ Hz. For the 650 nm laser, power $P = 66$ $\mu$W, beam waist $\omega = 22.9$ $\mu$m, $I_{650} = 75.5I_{sat}$, $\Delta_{D1} = 2\pi \times 349.9$ THz, scattering rate $1.29 \times 10^{-6}$ Hz. Therefore, both 493 nm and 650 nm laser beams provide the limitation of the coherence time around $4 \times 10^5$ s.

(v) Leakage of microwave: after improving the frequency stability of the local oscillator and suppressing magnetic-field fluctuations, the coherence time was improved to only twice, 1200 s, which was limited by the microwave leakage. We suppress the leakage by adding the microwave switch with 70 dB isolation at the final stage before the horn. We observe the enhancement of coherence time to 5400 s, which now is mainly limited by the frequency stability of the local oscillator as discussed in section (i). We estimate that the 70 dB isolation suppresses the carrier Rabi-frequency by microwave leakage around 3000 times, which improves the coherence-time limitation to around $4 \times 10^6$ ($\approx$1200 s $\times$ 3000).

(vi) Collision of background gas: background-gas collisions cause decoherence by collision frequency shift. The model in Ref. [52] estimates that $^{27}Al^+$ optical transition clock has a frequency shift of order $10^{-16}$ after 0.15 s probe from the background gas collision of $H_2$ in the pressure of 38 nPa at room temperature. The model estimates that a microwave transition has a larger shift as the level of $10^{-14}$ with 4 s probe, due to no suppression introduced by the Debye–Waller factor. This

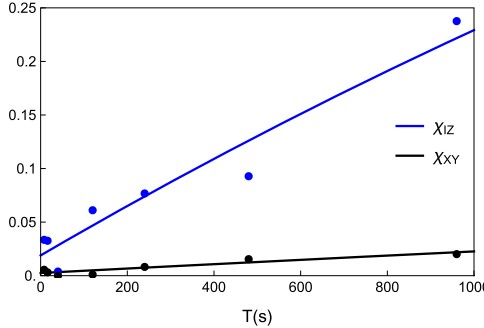

**Fig. 8 Fitting of process matrix elements evolution.** Blue and black points are experimental results of $\chi_{IZ}$ and $\chi_{XY}$, respectively. The solid lines are their fitting results.

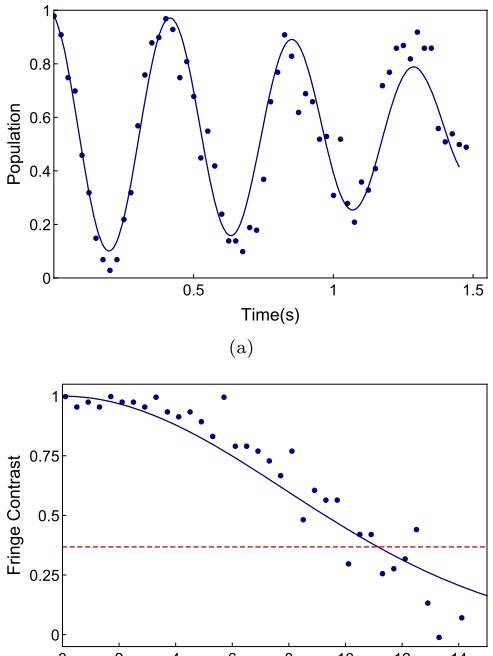

**Fig. 9 Coherence time obtained from Ramsey measurements. a** without dynamical decoupling pulses and (b) with one spin-echo pulse. Blue points are experimental results and each point repeats 100 times. The solid lines are their fitting by the Gaussian decay function. (a) A small detuning between microwave frequency and qubit energy splitting is used here to observe the oscillation signal rather than the fringe contrast signal to wipe out the frequency drift effect. The coherence time is 1.6 ± 0.22 s. **b** The red-dashed line indicates the $1/e$ threshold. The coherence time is 11.1 ± 0.38 s.

shift will be an upper bound of our collision frequency shift because the model does not include the suppression by the sympathetic cooling. We numerically simulate the collision frequency shift with $KDD_{xy}$ sequences. The infidelity of a superposition state is estimated by around $1.7 \times 10^{-9}$ for each $KDD_{xy}$ unit, which leads the coherence-time limitation to $4 \text{ s} \times 1/(1.7 \times 10^{-9}) \sim 2 \times 10^9$ s, where we assume the infidelity increase exponentially with the number of $KDD_{xy}$ gate numbers[59].

(vii) Lifetime of hyperfine state: The spontaneous emission rate of magnetic dipole transitions is written as $\gamma = \frac{\alpha \Delta_{HF}^3 |M|^2}{3 m_e^2 c^4}$, where $M$ is the magnetic dipole matrix element that is expected to be of order $\hbar$, $\alpha$ is the fine-structure constant, and $\Delta_{HF}$ the energy splitting of hyperfine qubit[60]. For the ground hyperfine level of $^{171}Yb^+$ ions, we estimate it as $\tau_{HF} = \frac{1}{\gamma} \sim 5 \times 10^{11}$ s, where we assume $M \sim \hbar$.

**Process matrix evolution**. We obtain the $T_1$ and $T_2$ in the diagonal elements of $\chi$ of Eq. (1) by fitting $\chi_{XY} \equiv 0.5(\chi_{22} + \chi_{33})$ and $\chi_{IZ} \equiv 1 - (\chi_{11} - \chi_{44})$ to the functions

of $\frac{1 - e^{-t/T_1}}{4}$ and $1 - e^{-t/T_2}$, respectively. As shown in Fig. 8, we obtain $T_1 = 11900 \pm 2200$ s and $T_2 = 4200 \pm 580$ s by fitting $\chi_{XY}$ and $\chi_{IZ}$, respectively. We note that ideally the total dephasing time $T_2 = 4200 \pm 570$ s in the process tomography should be matched to the coherence time of $5500 \pm 670$ s. The discrepancy originates from the quantum fluctuation noise in the other bases measurements of the process tomography. The process tomography requires measurements of four different bases for different input states. For example, a superposition input state, $(|0\rangle + |1\rangle)/\sqrt{2}$ (an eigenstate of $\sigma_x$), we need to measure the expectation values of identity, $\sigma_x$, $\sigma_y$, and $\sigma_z$. In principle, both $\langle \sigma_y \rangle$ and $\langle \sigma_z \rangle$ should be zero (even there exists serious decoherence). However, due to the quantum fluctuation noise, they deviated from zero, which introduced the reduction of the $T_2$ in the process tomography in our measurement. If these results are zero, the Ramsey coherence time and the total dephasing time of the process tomography will be perfectly matched. We believe if the number of measurements for the process tomography approaches infinity, the difference should converge to zero.

**Simple coherence time measurement**. Many experiments of interest can take advantage of dynamical decoupling pulses, but some of them cannot or can only apply a single spin-echo pulse. This makes the enhancement of these special-cases coherence time more attractive for some applications. Figure 9 shows the measurement results for direct Ramsey measurement and one spin-echo pause case.

## Data availability
The data that support the findings of this study are available from the corresponding author upon reasonable request.

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

## Acknowledgements

The authors thank Roee Ozeri, Rene Gerritsma, Jianwei Zhang, and Jize Han for helpful discussions. This work was supported by the National Key Research and Development Program of China under Grant Nos. 2016YFA0301900 and 2016YFA0301901, the National Natural Science Foundation of China Grant Noa. 11574002, 11974200, and 11504197, Singapore Ministry of Education through Tier 1 Grant No. RG190/17, the Singapore National Research Foundation through Fellowship No. NRF-NRFF2016-02, and NRF-ANR Grant No. NRF2017-NRF-ANR004 VanQuTe. X.Y. acknowledges the support from Simons Foundation.

## Author contributions

P.W., C.-Y.L., M.Q., M.U., J.Z., and Y.W. contributed to constructing the experimental system. P.W. with the assistance of C.-Y.L. performed the data taking and analysis. X.Y., M.G., and J.-N.Z provided theoretical support. K.K. supervised the experiment. All authors discussed the results and contributed to the writing of the manuscript.

## Competing interests

The authors declare no competing interests.

## Additional information

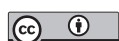

