## [Peer Review File · Nature Communications]

Review of “Single-ion qubit exceeding one hour coherence time” by P. Wang et al

This manuscript describes the demonstration of very long coherence times in a $^{171}\text{Yb}^+$ qubit (sympathetically cooled by a co-trapped $^{138}\text{Ba}^+$ ion). The authors have extended the already-long coherence times demonstrated in a previous paper through improved laser systems, the implementation of a lower-noise local oscillator, two-stage magnetic shielding, and a few other experimental upgrades. The authors demonstrate $\sim 80\%$ Ramsey contrast for experimental times of 900s, and extrapolate this to a $1/e$ coherence time of nearly 5500 s, longer than 1 hr. The authors also characterize the coherence via a number of other metrics and obtain broadly consistent results.

Long coherence times are important for numerous applications in quantum sensing and quantum information processing, and one of the chief advantages of trapped ions for these applications is their ability to achieve such long coherence times. This manuscript demonstrates the longest coherence time yet achieved in a trapped ion (by nearly an order of magnitude), comparable to the longest coherence times achieved in any quantum system. The key experimental advances necessary to achieve this result are described and the data are clearly presented and provide firm support for most of the claims. I believe this result will be of interest to specialists in the trapped-ion field and in fact to all researchers working in the area of quantum information processing. However, some areas of the manuscript are not clearly written and require improvement, and a few of the claims should be modified or better supported. If the authors can address these concerns, I believe the manuscript will merit publication in *Nature Communications*.

My concerns are summarized as follows.

1. The manuscript’s key claim is a coherence time of around 5500 s (over 1 hour). However, the longest measurement that was actually performed was only around 900s; claims of >1 hour coherence were extrapolated from much shorter times. I believe the manuscript’s central claim would be much stronger if at least one experiment that directly showed significant coherence remaining after 1 hour could be performed.

I understand that in the current circumstances (i.e., covid-19), it may not be feasible to take this additional data. I do not think that this data is absolutely essential to publication. However, if it cannot be taken in the present circumstances, I think the title/abstract should be modified to indicate that the achieved coherence times are extrapolations from shorter measurements.

2. Similarly, the abstract claims a coherence time of over 6000 s. However, of the many different measures of coherence used, only one gives a number over 6000 s. Moreover, the different measurements have error bars on the order of 600 s. Taking this into account, I think an average of the different measures suggests that the actual coherence is likely over 5000 s but not 6000.
3. The abstract is missing a key word: “order of improvement compared to the state-of-the-art record” should of course become “order-of-magnitude improvement.”
4. The >1 hr coherence time reported uses dynamical decoupling pulses to remove the effects of various slow perturbations in the lab. This is fair, and many experiments of interest can take advantage of these pulse sequences, but some cannot (i.e., optical clocks). It would be nice to also report the coherence that is achieved without the use of dynamical decoupling sequences,

if only so that researchers can understand what could reasonably be achieved in the best systems.

5. Many of the coherence times reported use too many significant figures given their error bars.
6. While it's true that the different coherence measures used largely agree, they do sometimes seem to differ by amounts larger than their error bars. For example, the Ramsey decoherence gives $T = 5500 \pm 670$ s, but for the process tomography, the overall T (given by $T^{-1} = 1/T_1 + 1/T_2$) seems like it would be only about 3000 s. The authors should discuss this discrepancy.
7. On p. 1, the statement "there is no theoretical limit of the coherence time of quantum systems," is not really true. The T_1 time is ultimately limited by the excited state lifetime, and $T_2 < 2 T_1$. For hyperfine qubits T_1 is extraordinarily long, but there is still some theoretical limit. For many other quantum systems the theoretical limit is not long at all! This should be restated and clarified.
8. The authors use a number of different metrics to quantify their coherence. It is useful to include more than one and show that they are generally consistent, but it's unclear why the authors have chosen to include so many (other than completeness). If one or more of these metrics are providing additional information on the nature of the decoherence processes that will be useful for the reader, this ought to be clearly stated. Otherwise I think that several of the more obscure ones could be relegated to the Supplemental Information (with a brief mention in the text that they were performed and yielded consistent results).
9. The descriptions of the later decoherence metrics are not at all clear (particularly RQM and REC) and should be improved. For example, for the metric RQM, the quantities N and M are not defined at all; it's very unclear how this is being performed. As mentioned earlier, the authors should also clarify why this metric gives additional information beyond what can be understood from the earlier metrics. If it's just being included for completeness it should be moved to the SM. Even if the key result remains in the main text, I think it is important that enough additional information be provided about these metrics (likely in the SM) so that the reader can understand what measurements were performed and what the results were. Figure 5 gives so little information to the reader that it currently does not contribute to the manuscript.
10. In general, the introduction, apparatus description, and initial Ramsey measurement description are well-written. (There are a few typos here and there which can hopefully be cleaned up). Later parts of the manuscript—particularly the alternative coherence measurements, conclusion, and Supplementary Material—are not as clearly written. There are some sentences which I cannot understand at all. The authors should go through these sections and make sure that they are written in comprehensible English. I've included a few examples below.
 - a. "The process matrix describes the quantum memory that in the beginning, no decoherence, and $t \gg T_1; T_2$ any initial states are changed to fully mixed state." This should be changed to "The process matrix describes a quantum memory with full coherence at $t = 0$ but which has transitioned to a fully mixed state for $t \gg \min(T_1, T_2)$."
 - b. The conclusions do not clearly indicate what the limitations to coherence would be under various conditions—for example, what would coherence time be expected to be in a room-temp system if the LO were improved? How about in a cryo system? The reader is referred to the SM but there is very little information there. One statement in the conclusion – "We find that hopping causes serious problems" – is directly

contradicted by a statement in the SM, “we do not observe any limitation from hopping problem.” This needs to be cleared up.

- c. The writing in the SM is very unclear and needs to be improved. I do not understand the statement, “The dynamical decoupling pulses with the interval of 0.4 s can compensate the frequency changes in about 10 minutes.” The part about magnetic shielding refers to “deeps in the data” when it means “dips.” Furthermore, the experiment performed for Fig. 6 in the SM is not described – I believe it looks at Ramsey contrast as a function of the inter-DD pulse spacing τ , but a sentence or two to describe this experiment is needed. The sections “Procedure of quantum process tomography of a single qubit” and “Quantum process evolution” need to be expanded—they are literally one sentence each and they provide no help to the reader at all. Figure 7 is likewise unexplained; what this figure is showing needs to be explained to the reader in at least a few sentences. (Among other things, the procedure by which T1 and T2 are derived from that data must be included).

Reviewer #2:

Remarks to the Author:

Dear editor,

In their manuscript "Single ion-qubit exceeding one hour coherence time", the authors Pengfei Wang et al. present results on characterizing the coherence time of a single $^{171}\text{Yb}^+$ trapped-ion hyperfine qubit. Using a combination of several technological components and methods, they achieve a coherence decay on the timescale of one hour. While this result is certainly impressive, this work is in large parts similar to Ref. 28, published by the same group. The key difference is the suppression of magnetic field fluctuations achieved by using a μ metal chamber and permanent magnets, as described in Ref. 33, which leads to an about six-fold improvement. The relation of this work to Ref. 28 is not even clearly described in the manuscript. Comparing to Ref. 28, I find the present work highly incremental and judge that it does not meet the impact criteria of Nature Communications.

Reviewer #1 (Remarks to the Author):

Comment. Long coherence times are important for numerous applications in quantum sensing and quantum information processing, and one of the chief advantages of trapped ions for these applications is their ability to achieve such long coherence times. This manuscript demonstrates the longest coherence time yet achieved in a trapped ion (by nearly an order of magnitude), comparable to the longest coherence times achieved in any quantum system. The key experimental advances necessary to achieve this result are described and the data are clearly presented and provide firm support for most of the claims. I believe this result will be of interest to specialists in the trapped-ion field and in fact to all researchers working in the area of quantum information processing. However, some areas of the manuscript are not clearly written and require improvement, and a few of the claims should be modified or better supported. If the authors can address these concerns, I believe the manuscript will merit publication in Nature Communications.

Our response: We greatly appreciate the referee’s full understanding of our works and constructive suggestions and comments to improve the manuscript. As shown below, we revise the manuscript comprehensively following the comments. After incorporating these revisions, we feel that the manuscript has been much more strengthened in its scope and clarity.

Comment 1. The manuscript’s key claim is a coherence time of around 5500 s (over 1 hour). However, the longest measurement that was actually performed was only around 900 s; claims of >1 hour coherence were extrapolated from much shorter times. I believe the manuscript’s central claim would be much stronger if at least one experiment that directly showed significant coherence remaining after 1 hour could be performed. I understand that in the current circumstances (i.e., covid-19), it may not be feasible to take this additional data. I do not think that this data is absolutely essential to publication. However, if it cannot be taken in the present circumstances, I think the title/abstract should be modified to indicate that the achieved coherence times are extrapolations from shorter measurements.

Our response: We agree with the referee. In the revised manuscript, as suggested by the referee, we modify the abstract to indicate that the achieved coherence times are extrapolations from shorter measurements as follows “Then, we observe the coherence time of around 5500 s for the $^{171}\text{Yb}^+$ ion-qubit, which is obtained by extrapolation from the measurements up to 960 s.”

Comment 2. Similarly, the abstract claims a coherence time of over 6000 s. However, of the many different measures of coherence used, only one gives a number over 6000 s. Moreover, the different measurements have error bars on the order of 600 s. Taking this into account, I think an average of the different measures suggests that the actual coherence is likely over 5000 s but not 6000 s.

Our response: In the abstract, the time of over 6000 s is not coherence time but the time for the robustness of quantum memory. In the revised abstract, we remove the time to avoid any confusion. We mention the coherence time of around 5500 s, which is obtained by Ramsey contrast that has been used most commonly in previous experimental demonstrations [13-17, 22-25, 28] for the fair comparison.

Comment 3. The abstract is missing a key word: “order of improvement com-

pared to the state-of-the-art record” should of course become “order-of-magnitude improvement.”

Our response: In the abstract, we decide to remove the expression of “order-of-magnitude” to address the comments by Referee 2. We correct related terms in the main text.

Comment 4. The >1 hr coherence time reported uses dynamical decoupling pulses to remove the effects of various slow perturbations in the lab. This is fair, and many experiments of interest can take advantage of these pulse sequences, but some cannot (i.e., optical clocks). It would be nice to also report the coherence that is achieved without the use of dynamical decouple if only so that researchers can understand what could reasonably be achieved in the best systems.

Our response: The data of a direct Ramsey measurement (that is, without the use of dynamical decouple) and a single spin-echo measurement are added in the Methods section. After all the improvements of suppressing magnetic-field noise and microwave leakages and using smaller Allan-variance frequency-reference, we observed around 1.6 s and 11.1 s coherence time for the Ramsey measurement without any dynamical decoupling pulse and with only a single spin-echo pulse.

Comment 5. Many of the coherence times reported use too many significant figures given their error bars.

Our response: We adjust the significant figures properly throughout the manuscript as suggested.

Comment 6. While it’s true that the different coherence measures used largely agree, they do sometimes seem to differ by amounts larger than their error bars. For example, the Ramsey decoherence gives $T = 5500 \pm 670$ s, but for the process tomography, the overall T (given by $T^{-1} = 1/T_1 + 1/T_2$) seems like it would be only about 3000 s. The authors should discuss this discrepancy.

Our response: We appreciate the referee’s point. As the referee correctly pointed out, the Ramsey measurements depending on times can be used to extract the total dephasing time, typically denoted as T_2 . In our process tomography, the time denoted as T_2 is the total dephasing time, not pure dephasing time (typically denoted as T_2^*). In our revised manuscript, we clearly state that T_2 is the total dephasing time for both the Ramey measurement and the process tomography.

For the total dephasing time, the Ramsey decoherence $T_2 = 5500 (\pm 670)$ s should be in agreement with $T_2 = 4200 (\pm 570)$ s in the process tomography within the error bars. They differ by amounts a little larger than error bars of standard deviation. We found the discrepancy originated from the quantum fluctuation noise in the other bases measurements of the process tomography. The process tomography requires measurements of four different bases for different input-states. For example, a superposition input-state, $(|0\rangle + |1\rangle)/\sqrt{2}$ (an eigenstate of σ_x), we need to measure identity, σ_x , σ_y , and σ_z . In principle, both $\langle \sigma_y \rangle$ and $\langle \sigma_z \rangle$ should be zero (even there exists serious decoherence), but due to the quantum fluctuation noise, they deviated from zero, which introduced the reduction of the T_2 in the process tomography in our measurement. If these results are zero, the Ramsey coherence time and the total dephasing time of the process tomography will be perfectly matched.

We believe if the number of measurements for the process tomography approaches infinity, the difference should converge to zero. We include the discussion in the Methods section.

Comment 7. On p. 1, the statement “there is no theoretical limit of the coherence time of quantum systems,” is not really true. The T_1 time is ultimately limited by the excited state lifetime, and $T_2 < 2 T_1$. For hyperfine qubits T_1 is extraordinarily long, but there is still some theoretical limit. For many other quantum systems the theoretical limit is not long at all! This should be restated and clarified.

Our response: We agree with the Referee. We revise the statement as “While the fundamental limit is far beyond 10 min,...”

Comment 8. The authors use a number of different metrics to quantify their coherence. It is useful to include more than one and show that they are generally consistent, but it’s unclear why the authors have chosen to include so many (other than completeness). If one or more of these metrics are providing additional information on the nature of the decoherence processes that will be useful for the reader, this ought to be clearly stated. Otherwise I think that several of the more obscure ones could be relegated to the Supplemental Information (with a brief mention in the text that they were performed and yielded consistent results).

Our response: We study a number of different metrics to quantify quantum coherence thanks to the full controllability of our quantum memory, which might not be accessible for ensemble systems or other limited systems. We relate our experimental results to not only traditional approaches but also recent rigorous theories of quantum coherence and quantum memory as quantum resources.

More specifically, in total, we include the five metrics as Ramsey contrast, process fidelity, mean fidelity, REC (Relative Entropy of Coherence) and RQM (Robustness of Quantum Memory). The Ramsey contrast has been used most commonly in previous experimental demonstrations. We include this result to fairly compare with other previous demonstrations. The quantum process tomography completely characterizes the decoherence process of our quantum memory. Therefore, it is the central part of the analysis of our quantum memory. However, the result of process tomography, the process matrix, may not be intuitive and simple to grasp the performance of the quantum memory. The mean fidelity can show the performance more intuitively. The performance of the quantum memory can be seen directly by examining the output fidelity of each initial state, which can be simplified to the mean fidelity. The mean fidelity averages out the fidelities of output states of the quantum memory of uniformly sampled initial states.

Recently due to the fundamental importance of quantum coherence, there have been active researches for rigorous theories of quantum coherence and quantum memory as a physical resource, which can be the standard quantifiers eventually. In our manuscript, we connect our experimental results to up-to-date resource theories of quantum coherence and quantum memory such as REC and RQM, respectively. The REC is a coherence quantifier based on distance measure [31], which was suggested as a gold standard measure [49]. The REC can be interpreted as the minimal amount of noise required for fully decohering the given state [31] and has the same formula with distillable coherence, which has an analogy to the distillable entanglement,

a standard widely-using entanglement quantifier. The distillable coherence is the optimal number of maximally coherent single-qubit states that can be obtained a given qubit state through incoherent operations [31]. It fulfills all the requirements as a proper coherence quantifier as suggested in Ref. [31].

The RQM [32] quantifies how well the memory preserves quantum information that includes coherence. Here the quantum memory, which stores a quantum state for later retrieval, is considered as a channel that maps an input state to an output state. Ideally, it should be an identity channel. The quantifier of RQM is developed based on the approach that considers the quantum memories as a resource and provides a means to benchmark quantum memories. Basically, the higher the RQM is, the more noise the quantum memory can sustain before it is unable to preserve quantum information. In contrast, a classical memory that cannot preserve quantum information is characterized as a measure-and-prepare (MP) memory that destroys the input state by measurement, and stores only the classical measurement result. The RQM is defined as the least portion of the classical memory that needs to be mixed with the quantum memory so that the resultant mixture belongs to MP memory, which is formally written as $R(\mathcal{N}) = \min_{M \in \mathcal{F}} \left\{ s \geq 0 \mid \frac{\mathcal{N} + s\mathcal{M}}{s+1} \in \mathcal{F} \right\}$, where \mathcal{N} is the quantum memory of interest, \mathcal{M} is a classical memory that is in the set of MP memories \mathcal{F} , and s is the amount of mixture of the quantum memory \mathcal{N} with the classical memory \mathcal{M} . The RQM is the minimum value of s to make the mixed memory is in \mathcal{F} .

We include the above discussions in the revised manuscript with proper section titles.

Comment 9. The descriptions of the later decoherence metrics are not at all clear (particularly RQM and REC) and should be improved. For example, for the metric RQM, the quantities N and M are not defined at all; it's very unclear how this is being performed. As mentioned earlier, the authors should also clarify why this metric gives additional information beyond what can be understood from the earlier metrics. If it's just being included for completeness it should be moved to the SM. Even if the key result remains in the main text, I think it is important that enough additional information be provided about these metrics (likely in the SM) so that the reader can understand what measurements were performed and what the results were. Figure 5 gives so little information to the reader that it currently does not contribute to the manuscript.

Our response: We improve the descriptions related to recently-developed rigorous coherence and memory quantifiers (REC and RQM, respectively) in the revised manuscript as also explained above. For the RQM, we clearly explain the definition of N and M and clarify the additional information of the RQM beyond the other metrics. We also include how to obtain the RQM, which is calculated from the process-tomography results. We include additional information that the REC and the RQM provide beyond the earlier metrics such as Ramsey contrast, process fidelity and mean fidelity with sufficient details of required measurements and results. Figure 5 shows only the experimental results and related discussions are included in the main text.

Comment 10. In general, the introduction, apparatus description, and initial Ram-

sey measurement description are well-written. (There are a few typos here and there which can hopefully be cleaned up). Later parts of the manuscript—particularly the alternative coherence measurements, conclusion, and Supplementary Material—are not as clearly written. There are some sentences which I cannot understand at all. The authors should go through these sections and make sure that they are written in comprehensible English.

Our response: As discussed in the response of comments 8 and 9, we comprehensively revise the sections of the alternative coherence measurements. We also include proper section titles and significantly revise these parts to make them much more clear.

Comment 10a. “The process matrix describes the quantum memory that in the beginning, no decoherence, and $t \gg T_1; T_2$ any initial states are changed to fully mixed state.” This should be changed to “The process matrix describes a quantum memory with full coherence at $t = 0$ but which has transitioned to a fully mixed state for $t \gg \min(T_1, T_2)$.”

Our response: We change the statement as suggested.

Comment 10b. The conclusions do not clearly indicate what the limitations to coherence would be under various conditions—for example, what would coherence time be expected to be in a room-temp system if the LO were improved? How about in a cryo system? The reader is referred to the SM but there is very little information there. One statement in the conclusion – “We find that hopping causes serious problems”—is directly contradicted by a statement in the SM, “we do not observe any limitation from hopping problem.” This needs to be cleared up.

Our response: We comprehensively revise the conclusion. The conclusion discusses the main extension of our current demonstration. The first is about how to further enhance the coherence time to the ultimate coherence time limited by the lifetime of the hyperfine state. The second is about how to increase the number of qubits in quantum memory. The hopping causes serious problems for multiple-qubit quantum memory but is not the limitation of current coherence time. In the revised version, we separate the discussions into two paragraphs to avoid any confusion.

Moreover, we identify the sources of coherence-time limitations and quantify how much limited the coherence time was because of them. The sources are categorized into the following 7 categories as (i) Phase noise of local oscillator; (ii) Magnetic-field fluctuation; (iii) Ion hopping; (iv) Scattering of $^{138}\text{Ba}^+$ lasers; (v) Leakage of microwave; (vi) Collision of background gas; (vii) Lifetime of hyperfine ground-state. According to our analysis, if the LO were improved, the next limitation comes from the magnetic field fluctuations. If both of the LO and field fluctuation are improved, the main limitation in the room temperature system comes from ion-hopping, which leads to the coherence-time limitation of 2×10^5 s (about two days). In the cryo system with a similar vacuum to room-temp system (10^{-10} torr), assuming all the other technical problems are infinitely suppressed, the limitation comes from the frequency shift by back-ground gas collision, which leads to the coherence-time limitation of 2×10^9 s (about 60 years). In the cryo system with 10^{-13} torr vacuum, the coherence time is limited to the fundamental limit, the lifetime of the ground hyperfine state.

Comment 10c. The writing in the SM is very unclear and needs to be improved. I do not understand the statement, “The dynamical decoupling pulses with the interval of 0.4 s can compensate the frequency changes in about 10 minutes.” The part about magnetic shielding refers to “dips in the data” when it means “dips.” Furthermore, the experiment performed for Fig. 6 in the SM is not described – I believe it looks at Ramsey contrast as a function of the inter-DD pulse spacing τ , but a sentence or two to describe this experiment is needed. The sections “Procedure of quantum process tomography of a single qubit” and “Quantum process evolution” need to be expanded—they are literally one sentence each and they provide no help to the reader at all. Figure 7 is likewise unexplained; what this figure is showing needs to be explained to the reader in at least a few sentences. (Among other things, the procedure by which T_1 and T_2 are derived from that data must be included).

Our response: We entirely reorganize the SM (now Methods), include the details of the coherence-time limitations by the 7 sources with summary figure (Fig. 6), provide the detailed experimental procedure for Fig. 6 and Fig. 7 (now Fig. 7 and Fig. 8, respectively), and expand the section of “Quantum process evolution.” The section of “Procedure of quantum process tomography of a single qubit” is combined with the main text. We believe the revised Methods section is clear and accessible to a broad readership.

Reviewer #2 (Remarks to the Author):

Comments: In their manuscript “Single ion-qubit exceeding one hour coherence time”, the authors Pengfei Wang et al. present results on characterizing the coherence time of a single $^{171}\text{Yb}^+$ trapped-ion hyperfine qubit. Using a combination of several technological components and methods, they achieve a coherence decay on the timescale of one hour. While this result is certainly impressive, this work is in large parts similar to Ref. 28, published by the same group. The key difference is the suppression of magnetic field fluctuations achieved by using a mu metal chamber and permanent magnets, as described in Ref. 33, which leads to an about six-fold improvement. The relation of this work to Ref. 28 is not even clearly described in the manuscript. Comparing to Ref. 28, I find the present work highly incremental and judge that it does not meet the impact criteria of Nature Communications.

Our response: We thank the referee for acknowledging that our result is impressive, which is about a nine-fold enhancement of coherence time from 10 min to 90 min. As we clearly state in the abstract and main text, such enhancement of the coherence time was achieved by the following three technical advancements: the suppression of magnetic field fluctuation, the improvement of the reference clock for microwave oscillator, and the reduction of microwave leakage. It is not improved by only the suppression of magnetic field fluctuations achieved by using a mu-metal chamber and permanent magnets.

At the time when we achieved a coherence time of 10 min, we also naively assumed that the coherence time was limited by the magnetic field fluctuation. After we installed the mu-metal shielding to enclose our vacuum system with the ion trap, surprisingly, we did not observe any increase of the coherence time of the clock-state qubit, though we observed over 30 times longer coherence time for the Zeeman qubit. Therefore, we can surely say that just the suppression of field fluctuation is not enough to enhance the coherence time. After desperately struggling, we found that the limitation of the coherence time of 10 min did not originate from the qubit itself, but frequency instability of the microwave oscillator and leakage of the microwave. Recently, the importance of frequency stability of reference oscillators also was discussed in Ref. [Phys. Rev. Lett. 123, 110503 (2019)]. Even after using a frequency reference with an order-of-magnitude smaller Allan-variance, the coherence time was increased to 1200 s, about twice improvement. Finally, after we suppressed the leakage of the microwave, we were able to observe the coherence time of 90 min, which is mainly limited by the instability of the microwave oscillator to our serious analysis included in the revised manuscript.

In this revised manuscript, we more explicitly state these three main technical improvements in the abstract, and include the subsections with titles of these technical improvements in the main text. We also made a more clear connection of this work to our previous work of Ref. 28 in the abstract and main text. Furthermore, with the capability of full control on the single-qubit of long coherence time, we systematically studied the process of decoherence and made the connection to recently developed coherence theory as a fundamental quantum resource. Our experimental demonstration of a long coherence time of single-qubit will accelerate the realization of practical quantum applications such as quantum money, which ultimately requires infinite coherence time.

In general, the authors have addressed the concerns I raised previously. I especially commend the authors for making the later sections of the manuscript much more clear and comprehensible. I still have a number of remaining comments that I hope will be addressed prior to publication. I continue to believe that this manuscript merits publication in *Nature Communications*, and I believe the article will be of interest to many researchers.

- While the writing is mostly clear, there are a number of places which require copy-editing. There are many missing articles (“the”, “a”, etc), nouns and verbs are sometimes missing (“a two-layer of μ -metal shielding...”, “lifetime...that is expected to thousands of years”, etc), and there are various other grammatical problems. I hope the authors will work closely with the editors to improve the grammar; pointing out all of these places seems beyond the scope of the review.
- In the abstract, the discussion of quantum money is kind of unexpected, and other quantum information areas should be mentioned as well. Moreover, if quantum money needs “unlimited storage time,” it’s unachievable and the advance reported in this paper does not actually bring you closer to an unlimited time.
- Moreover, there are many other applications for long coherence time, especially in the noisy-intermediate-scale quantum (NISQ) regime where quantum information processors will not have error correction! The abstract and Discussion sections should make this a little clearer—the emphasis on quantum money is a little strange as I don’t think it is considered the most promising or important quantum tool.
- When insets in the paper show extrapolation of fits to very long times (fig 3, fig 4, etc), the captions should clearly identify that these are extrapolations of fits.
- The discussion of previous work on p.1 first column is not very clear, the sequence of improved times and technical improvements is generally hard to follow and should be improved. For the sentence “The main limitation comes from the problem of qubit-detection inefficiency due to motional heating of qubit-ions without Doppler laser-cooling. The limitation was addressed by sympathetic cooling by other species of ion...” The first “main limitation” should be put in the past tense, such as “A main limitation due to motional heating of qubit-ions without Doppler cooling was addressed by sympathetic cooling of other ion species...”. This sentence is describing previous work and treating it as a current limitation is very confusing.
- On p. 2 beginning of second column, rewrite: “which has an order-of-magnitude smaller Allan variance at 1 s observation time than our previous Rb clock oscillator.”
- On p. 4 bottom of first column, the paragraph on “performance of the quantum memory on arbitrary quantum states” is still confusing. It should be made clear that the mean fidelity is being obtained as a function of wait time. The last sentence in this paragraph needs to be fixed, also the difference between the two times that were obtained here (5200 s and 5600 s) is not clear.
- In the Discussion, the claim “Our research can lead to general-purpose quantum-memory such as quantum money...” isn’t really phrased correctly. Quantum money is a specific application of a quantum memory, not a general one, and possibly not the most interesting. Again, there are many applications for very long coherence times.

Reviewer #1 (Remarks to the Author):

Review of “Single-ion qubit exceeding one hour coherence time” by P. Wang et al

Comment 1. In general, the authors have addressed the concerns I raised previously. I especially commend the authors for making the later sections of the manuscript much more clear and comprehensible. I still have a number of remaining comments that I hope will be addressed prior to publication. I continue to believe that this manuscript merits publication in Nature Communications, and I believe the article will be of interest to many researchers.

Our response: We thank the referee for recommending our manuscript for the publication of Nature Communications. We believe this revised version of the manuscript addressed all the remaining comments of the referee.

Comment 2. While the writing is mostly clear, there are a number of places which require copy-editing. There are many missing articles (“the”, “a”, etc), nouns and verbs are sometimes missing (“a two-layer of μ -metal shielding...”, “lifetime...that is expected to thousands of years”, etc), and there are various other grammatical problems. I hope the authors will work closely with the editors to improve the grammar; pointing out all of these places seems beyond the scope of the review.

Our response: We checked and corrected the grammatical errors throughout the manuscript.

Comment 3. In the abstract, the discussion of quantum money is kind of unexpected, and other quantum information areas should be mentioned as well. Moreover, if quantum money needs “unlimited storage time,” it’s unachievable and the advance reported in this paper does not actually bring you closer to an unlimited time.

Comment 4. Moreover, there are many other applications for long coherence time, especially in the noisy-intermediate-scale quantum (NISQ) regime where quantum information processors will not have error correction! The abstract and Discussion sections should make this a little clearer—the emphasis on quantum money is a little strange as I don’t think it is considered the most promising or important quantum tool.

Our response: We agree with the referee. There are many applications for long coherence time, especially in the NISQ regime. In this revision, we revised the abstract and the Discussion section to make this clearer.

Comment 5. When insets in the paper show extrapolation of fits to very long times (fig 3, fig 4, etc), the captions should clearly identify that these are extrapolations of fits.

Our response: In the revised captions of fig 3, fig 4, and fig 5, we clearly identify that these insets are extrapolations of fits as “Inset shows extrapolations of fits in a longer time range”.

Comment 6. The discussion of previous work on p.1 first column is not very clear, the sequence of improved times and technical improvements is generally hard to follow and should be improved. For the sentence “The main limitation comes from the problem of qubit-detection inefficiency due to motional heating of qubit-ions without Doppler laser-cooling. The limitation was addressed by sympathetic cooling by other species of ion...” The first “main limitation” should be put in the past tense, such as “A main limitation due to motional heating of qubit-ions without Doppler cooling was addressed by sympathetic cooling of other ion species...”. This sentence is describing previous work and treating it as a current limitation is very confusing.

Our response: We improved the discussion of previous works to make it easy to follow. Following the suggestion of the referee, we used the past tense for the works already done before our current work.

Comment 7. On p. 2 beginning of second column, rewrite: “which has an order-of-magnitude smaller Allan variance at 1 s observation time than our previous Rb clock oscillator.”

Our response: We changed the statement as suggested.

Comment 8. On p. 4 bottom of first column, the paragraph on “performance of the quantum memory on arbitrary quantum states” is still confusing. It should be made clear that the mean fidelity is being obtained as a function of wait time. The last sentence in this paragraph needs to be fixed, also the difference between the two times that were obtained here (5200 s and 5600 s) is not clear.

Our response: We modified the discussion of mean fidelity to make it more clear, and clearly say that the mean fidelity is a function of wait time. The difference between two times (5200 s and 5600 s) was included to show the consistency of our analysis. We revised the last part of the paragraph to clearly reveal the point.

Comment 9. In the Discussion, the claim “Our research can lead to general-purpose quantum-memory such as quantum money...” isn’t really phrased correctly. Quantum money is a specific application of a quantum memory, not a general one, and possibly not the most interesting. Again, there are many applications for very long coherence times.

Our response: As also raised in the Comments 3 and 4, we revised the discussion section as suggested. We included many general applications of the long coherence time as quantum computation, quantum communication, and quantum metrology.